# Identification of Putative Vaccine and Drug Targets against the Methicillin-Resistant *Staphylococcus aureus* by Reverse Vaccinology and Subtractive Genomics Approaches

**DOI:** 10.3390/molecules27072083

**Published:** 2022-03-24

**Authors:** Romen Singh Naorem, Bandana Devi Pangabam, Sudipta Sankar Bora, Gunajit Goswami, Madhumita Barooah, Dibya Jyoti Hazarika, Csaba Fekete

**Affiliations:** 1Department of General and Environmental Microbiology, Institute of Biology and Sport Biology, University of Pécs, Ifusag utja. 6, 7624 Pecs, Hungary; romen86@gamma.ttk.pte.hu (R.S.N.); bandanapangabam12@gmail.com (B.D.P.); 2Department of Agricultural Biotechnology, Assam Agricultural University, Jorhat 785013, India; madhumita.barooah@aau.ac.in (M.B.); dshine22@gmail.com (D.J.H.); 3DBT—North East Centre for Agricultural Biotechnology (DBT-AAU Center), Assam Agricultural University, Jorhat 785013, India; sudip.asm@gmail.com; 4Multidisciplinary Research Unit, Jorhat Medical College and Hospital, Jorhat 785008, India; gunajitgoswami1@gmail.com

**Keywords:** methicillin-resistant *Staphylococcus aureus*, core genome, reverse vaccinology, subtractive genome approach, drug target, vaccine candidate, molecular docking

## Abstract

Methicillin-resistant *Staphylococcus aureus* (MRSA) is an opportunistic pathogen and responsible for causing life-threatening infections. The emergence of hypervirulent and multidrug-resistant (MDR) *S. aureus* strains led to challenging issues in antibiotic therapy. Consequently, the morbidity and mortality rates caused by *S. aureus* infections have a substantial impact on health concerns. The current worldwide prevalence of MRSA infections highlights the need for long-lasting preventive measures and strategies. Unfortunately, effective measures are limited. In this study, we focus on the identification of vaccine candidates and drug target proteins against the 16 strains of MRSA using reverse vaccinology and subtractive genomics approaches. Using the reverse vaccinology approach, 4 putative antigenic proteins were identified; among these, PrsA and EssA proteins were found to be more promising vaccine candidates. We applied a molecular docking approach of selected 8 drug target proteins with the drug-like molecules, revealing that the ZINC4235426 as potential drug molecule with favorable interactions with the target active site residues of 5 drug target proteins *viz.*, biotin protein ligase, HPr kinase/phosphorylase, thymidylate kinase, UDP-N-acetylmuramoyl-L-alanyl-D-glutamate-L-lysine ligase, and pantothenate synthetase. Thus, the identified proteins can be used for further rational drug or vaccine design to identify novel therapeutic agents for the treatment of multidrug-resistant staphylococcal infection.

## 1. Introduction

Methicillin-resistant *Staphylococcus aureus* is a major human and animal pathogen, leading to cause infections ranging from skin and soft tissue infections (SSTI) to life-threatening infections such as endocarditis, bacteremia, and osteomyelitis [1,2,3,4]. The genomic plasticity of MRSA has facilitated the acquisition of antibiotic resistance genes (ARGs) and virulence factors-encoding genes (VFGs) that led to the emergence of multidrug-resistant (MDR) and hypervirulent MRSA. The emergence of the MDR strain has led to highly challenging issues in antibiotic therapy [5]. Consequently, the morbidity and mortality rates caused by MRSA infections have a substantial impact on health concerns [6]. The mortality rates of MRSA infections are higher than the combined mortality rates of HIV/AIDS, Parkinson’s disease, emphysema, and homicide [7,8]. Thus, the prevention and control of multidrug-resistant MRSA infections have become the main concern in the public health sector [9,10]. The treatment of MRSA infections generally relies on conventional antibiotic therapy, even though the number of MDR cases has been increasing rapidly. Conventional drug discovery methods are laborious, expensive, time-consuming, and often find few drug targets or lead compounds [11,12]. This strongly suggests that there is a continuous need to search for additional drug or vaccine molecules in their genomes that would improve long-lasting protection [13]. With the advancement in genome sequencing technologies, the number of bacterial genome sequences has increased rapidly, which provides an excellent opportunity to accelerate the drug target or vaccine candidate discovery process through computational approaches such as comparative and subtractive genomics, and reverse vaccinology [14,15]. These approaches enable an evaluation of bacterial proteins that can bind to drug molecules or induce an adaptive immune response [16]. The surface-exposed proteins mediate the initial process of pathogen attachment and evoke immune responses, regarded as suitable vaccine candidates [17,18]. This in silico technique optimizes the prediction of drug target and vaccine candidate proteins of different pathogens that are challenging to culture in the laboratory. Recently, such computational approaches had popular results for the rapid discovery of novel targets in various human pathogens including SARS-CoV-2 [19,20], Ebola virus [21], *Mycobacterium tuberculosis* [22], *Helicobacter pylori* [23], *Salmonella typhi* [24], serogroup B *Neisseria meningitides* [25], *Streptococcus agalactiae* [26], and *Acinetobacter baumannii* [27]. These approaches allowed to fill the gap between genomic data and the identification of 3D structures of therapeutic targets. The structural information from high-throughput comparative modeling for large-scale proteomics data for inhibitor identification potentially leads to the discovery of compounds able to prevent the survival of pathogens. These proteins, which must show no cross-reaction with the host, have been proposed as potential drug targets. This study would be extremely significant since MRSA drugs could prevent the deaths and burden of hospital charges. 

In this present study, the genomes of MRSA strains were studied using subtractive genomics and reverse vaccinology approaches to identify the potential drug targets and vaccine candidates. The overall aim was to identify drug molecules that show favorable interactions, with low energy values, and high complementarity with the predicted targets for global MRSA strains.

## 2. Results and Discussion

This study was carried out to identify the putative drug targets and vaccine candidates against MRSA strains for proposing alternative prospective therapeutics. Earlier, the development of new antimicrobial agents and vaccine therapies was limited due to a computational technology bottleneck. However, in this post-genomic era, advances in the fields of genomics, proteomics, coupled with the development of bioinformatics tools have allowed for in silico identification of new drug candidates and vaccine targets from genomics, and protein sequence resources using the subtractive genomic approach or reverse vaccinology approach [28]. The reverse vaccinology approach is a powerful method for the identification of unique yet uncharacterized sequences as possible therapeutic targets [29,30,31]. Reverse vaccinology reduces the time needed to develop new vaccines and offers flexibility even for non-cultivable pathogens [32]. This approach also allows not only the detection of all the antigens as observed through the conventional methods but also the discovery of novel antigens that function on a different paradigm [18].

### 2.1. Prediction of Core-Genome

A set of 1719 coding DNA sequences (CDSs) shared by all the strains were identified and these sequences were extracted in the form of protein sequences and considered as core proteome (Figure 1). Out of this core proteome, 1678 non-redundant protein sequences were retrieved by using CD-HIT with a tolerance threshold set at 70% (Appendix A).

### 2.2. Identification of Essential Proteins and Non-Homologous Proteins

The non-redundant protein sequences of the core proteome have large numbers of genes that are not essential for the survival of an organism [33]. Essential proteins show potential targets for drug designing, because the mainstream of antibacterial compounds is synthesized to dock essential proteins. This step primarily mines essential proteins of MRSA strains that are necessary for their survival within the host. Such proteins are housekeeping in nature and important for basic cellular functions [34]. When these essential proteins are confirmed to be virulent, they can be of vital significance to unveil novel therapeutic targets [35]. The essential core proteome that has homology among all the *S. aureus* strains has the potential to serve as a universal drug/vaccine target towards combating multidrug-resistant *S. aureus* strains [36]. The GEPTOP 2.0 server identified a total of 278 essential core proteins (Figure 1, Appendix A). Further, comparison of the core proteins of essential protein sequences to the human host proteome resulted in a set of 98 targets as essential non-host homologous (Appendix A), while a set of 184 targets as essential host homologous proteins (Appendix A). Proteins that are non-homologous and essential for pathogens are integral to developing species-specific drug targets [37,38], and this subtractive proteomic or genomic approach is very essential to avoid drug cross-binding with the human host proteins and the possibility of the drug inducing adverse effects [39].

### 2.3. Characterization and Prediction of Subcellular Location of Proteins

The conserved non-host homologous proteins of MRSA strains can localize at different regions including cytoplasmic, membrane, putative surface-exposed, and secretory. Therefore, the localization of proteins is an important aspect of designing any therapeutic agents such as drug targets or vaccine candidates [40]. Out of the 98 subcellular localized and essential non-homologous proteins, 78 proteins were localized in cytoplasmic (CYT), 2 proteins were secretory (SEC), 6 proteins were potentially surface-exposed (PSE), and 12 proteins were membrane (MEM)-bound (Figure 1, Appendix A). The proteins localized in cytoplasmic regions play a pivotal role in maintaining cell viability, and therefore, these proteins are considered drug targets [41]. In the case of vaccine candidates, the exposed proteins including membrane, putative surface-exposed, and secreted proteins are suitable for reverse vaccinology [14]. The membrane proteins have more priority for vaccine candidates due to their closer contact with the human host and initiating a cascade of immune responses. However, the proteins with more helices are difficult to purify from the bacteria, thus membrane proteins containing fewer helices (≤2) are more preferred [42].

### 2.4. Potential Vaccine Target Candidates

In antigen-based vaccine design, adhesins are key molecules for the design of vaccines against microbial infective agents [43]. With the advancement of bioinformatics, screening of adhesion molecules in the bacterial genome has become quicker through homology-based approaches. SPAAN (software program for the prediction of adhesins and adhesin-like proteins using a neural network) program integrated with Vaxign v.2.0 tool is based on highly curated datasets and neural networks that were optimally trained for compositional attributes. The probability for a given protein likely to be an adhesin is the weighted average of individual probabilities emerging from the five networks, based on the accuracy of each network [44]. The search for candidate adhesion proteins can be a most promising approach for identifying novel vaccine candidates [28]. 

The previous study on MRSA 252 genome revealed that IsaA, HlgA, SsaA, and IsdB were the best immunogenic targets [45]. In the present study, the 20 proteins were localized in secreted (SEC), surface-exposed (PSE), and membrane (MEM) regions. Among these proteins, Vaxign v.2.0 identified the 4 best putative vaccine candidates *viz.*, foldase protein, ESAT-6 machinery protein, penicillin-binding protein (PBP) 1, and PBP2. 

Foldase protein (PrsA) (WP_000782119.1) was found to be the most promising vaccine candidate among other candidates since this membrane protein has an accessory role in virulence [46]. This protein has high antigenicity (0.7662, and 0.7121 scores defined by VaxiJen, and ANTIGENPro, respectively), a great adherence score (0.68), low molecular weight (35.623 kDa), no transmembrane helix, and 5 B-cell epitope peptides (Table 1). PrsA has a PPIC-type PPIASE domain belonging to the Rotamase family (PF00639), this domain region is in 146–245 amino acid (aa) positions of the protein sequence, and acts as interconversion of *cis*-proline and *trans*-proline. It was reported that PrsA is a surface-exposed protein that is essential for protein folding and is involved in cell wall biosynthesis as well as bacterial pathogenicity [47,48]. This protein stimulates an antibody response and extends the protection against multiple mouse infections [49,50]. In *Streptococcus sanguinis*, this protein is reported to be the vaccine candidate with the induction of opsonic antibodies [51]. Further, PrsA is highly conserved in *Legionella pneumophila* and is significantly immunogenic, thus offering the scope for the development of a DNA vaccine [52]. Our study suggested that PrsA from MRSA could protect the different lineages of pathogenic *S. aureus* strains and may be a novel protective antigen for MRSA vaccine development. 

The ESAT-6 machinery protein (EssA) protein is the conserved membrane protein that is necessary for the synthesis and secretion of EsxA protein [53]. Secretion of EsxA was prevented in the absence of the *essA* gene, and this protein might play a role in the process of the pathogenesis for *S. aureus*. Previous studies indicated that EsxA was the important candidate antigen for *S. aureus* vaccine development [54]. An immune protective antigen, EssA is a highly homologous protein of *Mycobacterium tuberculosis* and has good virulence and immunogenicity [55]. EssA protein can induce a high level of an immune response against *Streptococcus agalactiae* infection [52]. In this study, EssA protein (WP_000928935.1) of *S. aureus* was found with a T7SS_EssA_Firm domain (PF10661) located at 2–144 aa positions in its sequence. This protein was found to bear MHC adhesion capacity with a score of 0.58; high antigenicity (0.7034, and 0.8367 scores defined by VaxiJen, and ANTIGENPro, respectively), and one B-cell epitope peptide (Table 1 and Appendix A). 

Every core genome of *S. aureus* possesses four penicillin-binding proteins (PBP1-PBP4) linked to peptidoglycan biosynthesis [56]. Among these PBPs, the first two, i.e., PBP1 and PBP2, play important roles in *S. aureus* survival [57]. Earlier studies reported that PBP1 and PBP2 are considered potential therapeutic targets for their cellular importance. These proteins were found interacting with the staphylococcal femXAB family protein, mur-family proteins, Ddl, and MraY and play a curial role in the peptidoglycan biosynthesis pathway [57,58]. Also, these proteins have immunogenic properties and could be vaccine candidates in *N. meningitidis* [59]. These PBP1 and PBP2 have the highest antigenicity scores of 0.9481, and 0.9454, respectively, defined by ANTIGENPro. It was reported that PBP1 from *M. tuberculosis* can be used to design a new TB vaccine [59]. Vaccination with PBP2 induces protection against a protein involved in chromosome-mediated antibiotic resistance for meningococcal disease [35,60]. However, the information on these two proteins for the vaccine development against *S. aureus* is scantly available. This study showed that PBP1 and PBP2 proteins have the good characteristics of antigenic values, adhesion scores, stability, and epitope numbers (Table 1 and Appendix A). PBP1 protein has three conserved domains, FtsI (COG0768), PASTA_PBP2x-like_1 (cd06576), and PASTA_PBP2x-like_2 (cd06575). FtsI domain is found in 28–592 regions of the protein sequence, controlling cell cycle, cell division, chromosome partitioning, and cell wall or membrane biogenesis. PASTA domains of PBP2x-like_1, and PBP2x-like_2 are found in the 601–655, and 659–712 aa regions of PBP1 protein sequence. These domains catalyze the peptidoglycan synthesis, which is essential for cell division and protects from osmotic shock and lysis. PBP2 is a DD-transpeptidase essential for bacterial cell wall synthesis. This protein has a conserved domain—MrcB (membrane carboxypeptidase B), which is found in the 46–727 aa regions of the sequence which bind to the β-lactamthiazolidine ring system of β-lactam antibiotics.

The identified vaccine candidate proteins are localized in MEM regions and have high stability, antigenic and non-allergen properties (Table 1). Also, these proteins have an adhesion score > 0.51, ensuring that these can efficiently bind on MHC class I and II molecules, and may induce either cellular or humoral adaptive immune responses [46]. It was reported that clinical trials to design effective anti-Staphylococcal vaccines have not been fruitful [61], thus numerous vaccines were reported that mainly focus to trigger B-cell response and development of antibodies opsonization [62]. Additionally, the presence of antigenic B-cell epitopes in these candidate proteins confirms the ability to interact with the MHC class I molecule. The interaction of MHC class I-presented antigens with cytotoxic CD8+ lymphocytes is one of the potential vaccine-induced immune responses [63]. The prospective candidate proteins reported in this study fulfill all the prerequisites (subcellular localization, antigenic and adhesin properties) of being potent vaccine candidates. Among the identified vaccine candidates, PrsA and EssA proteins have higher antigenic properties defined by VaxiJen than PBP1 and PBP2 proteins (Table 1). Previous studies suggested that the protein with the highest antigenicity could be recognized by immune response easily and evoke the immune responses [44,48,53,59,64]; therefore, PrsA and EssA proteins could be effective vaccine candidates. Further, the multiple epitopes from the identified vaccine candidates having different pathways could be used to develop a universal anti-staphylococcal vaccine [65].

### 2.5. High-Throughput Structural Modelling and Druggability Analysis

The CYT protein sequences were selected for drug target analysis using MHOLline 2.0, an online web tool, to predict the modelome. In this analysis, proteins that belonged to “very high”, “high”, and “good” structural qualities of the G2 model group were considered for further analysis. The G2 model predicted 19 proteins (9 very high quality, 8 high, and 2 good) as potential candidates for drug targets. The other factors involved in drug target prioritization are low molecular weight, and high druggability [66]. The ability of a protein to hold a pocket for the binding of small molecules is one of the key steps in the identification of a drug target [67]. Therefore, pocket druggability analyses are crucial in therapeutic drug discovery. In this study, we chose the 8 best potential drug target candidates based on the drug score and binding site residues defined by the DoGSiteScorer tool, and a molecular weight criterion of less than 90 kDa (Appendix A).

### 2.6. Virtual Screening and Molecular Docking Analysis

The ligands, drug-like molecules, were screened for favorable interactions with each target protein. The top compounds were further used for flexible docking analysis with the residues of most druggable cavities defined by the DoGSiteScorer web tool and MVD software. As a result, the predicted protein–ligand interactions with the active site residues of each target are represented in Table 2, with ZINC ID, MolDock score for the selected ligand as well as hydrogen (H)-bonds involved in the interaction. In molecular docking, lower energy scores represent better protein–ligand bindings compared to higher energy values [68,69]. Hydrophobic interactions are the major contributors to the stability of proteins. H-bonding also maintains protein stability, but to a lower extent than hydrophobic interactions, even in the smallest globular proteins. Accordingly, hydrophobic binding of a ligand to essential amino acid residues of protein is the main determinant of folding configuration equilibria in many native proteins [70]. Additionally, the Simplified Molecular Input Line Entry System (SMILES) format of drug molecules was analyzed for their ADME/pharmacokinetic profile and drug-likeness parameters using the swissADME web server [71].

#### 2.6.1. Biotin Protein Ligase

Biotin protein ligase (BPL) encoded by the *birA* gene plays are involved in biotin homeostasis by biotinylation, and transcription repressor activities [72]. This enzyme is the master regulator of all biotin-mediated metabolic processes in *S. aureus* and is an emerging new drug target [72,73,74]. It is also reported that this enzyme is essential for fatty acid biosynthesis and the tricarboxylic acid cycle pathways in the bacterium [75]. The crystallographic structure of the BPL template (PDB ID: 6NDL from *S. aureus*) has two crystallographic native ligands, where BQX (1-[4-(6-aminopurin-9-yl)butylsulfamoyl]-3-[4-[(4~{S})-2-oxidanylidene-1,3,3~{a},4,6,6~{a}-hexahydrothieno[3,4-d]imidazol-4-yl]butyl]urea) native crystallographic ligand participating H-bond interaction on active site residues of its template were Asp180, Arg125, Ser128, Arg120, Arg122, Lys187, Gln116, Ser93, Asn212, and Thr94. In this study, cavity 1 defined by MVD tool having Volume (V):160.768 Å; Surface (S):528.64 Å; Radius (R):20 Å of BPL template was docked with ZINC4235426 compound, which resulted in the formation of seven H-bonds with the active site residues, Try182, Arg227, Arg125, and Arg122, and showed a MolDock score of -176.846 (Figure 2a,b). Also, the ZINC4237101 compound formed no H-bonds with the active site residues of the BPL template and revealed the MolDock score of −167.239. 

The redocking of the co-crystallized structure of native ligand with its protein was an essential step to confirm that ligand bind within the binding pocket/cavity in the appropriate conformation, also the generated RMSD values of less than 2 Å is considered for docking accuracy [69]. The redocking of native ligand (BQX) with BPL (PDB ID: 6NDL) generated the MolDock score of −177.821, and RMSD (root mean square deviation) value of 1.8 Å, which indicated that the applied protocol is favorable for docking simulation.

#### 2.6.2. HPr Kinase/Phosphorylase

HPr kinase/phosphorylase is a bifunctional enzyme that enhances the glycolytic intermediates (carbon metabolism) and virulence progression in Gram-positive bacteria [76,77]. In *Listeria monocytogenes*, the metabolism of carbon sources inhibits the PrfA, a transcription activator, and is involved in the virulence gene expression regulation [76]. This enzyme is of clinical interest due to its regulatory roles in the infectious process and therefore could be a new drug target. The template crystal structure of HPr kinase/P (1KO7 from *S. xylosus*) interacted with native crystallographic ligand PO_4_. The cavity 1 (S:62.976 Å; V:198.4 Å; R:8 Å) of Hpr Kinase/P defined by the MVD tool was selected for docking. The ZINC4235426 compound formed four H-bonds with the active site residues, Lys259, Thr150, and Asn227, and generated the MolDock score of −147.451 (Figure 3a,b). Also, the ZINC4235924 compound formed seven H-bonds with the active site residues, Gly151, Thr150, Asn227, Lys258, Thr260, and Asn229 of HPr kinase/P, and revealed the MolDock score of −137.549 (Figure 3c,d). 

#### 2.6.3. Thymidylate Kinase (TMK)

Thymidylate kinase (TMK) is a nucleotide kinase that catalyzes deoxythymidine monophosphate to deoxythymidine diphosphate using adenosine triphosphate (ATP) as the source of the phosphoryl group and results in the biosynthesis of deoxythymidine triphosphate (dTTP) for DNA synthesis [78]. Therefore, TMK is considered an attractive potential target for antibacterial drug inhibition [79]. The crystal structure of TMK protein (4HLC from *S. aureus* subsp. *aureus* MRSA252) has active site residues (Gln101, Arg70, Arg48, and Ser97) that are involved in H-bond interactions with native ligand benzoic acid (T05). Although, none of these residues were predicted to form H-bonds with the ZINC4259578 compound. However, this compound was predicted to make three H-bonds with other active site residues, Arg75, Arg97, and Arg110 of TMK protein, and carried a MolDock score of −139.656 (Figure 4a,b). Also, the ZINC4235426 compound created six H-bonds with the active site residues, Arg75, Arg97, Glu106, Tyr105, and Glu42 of TMK protein with a predicted MolDock score of −139.150. The redocking of native ligand (T05) with TMK (PDB ID: 4HLC) generated the MolDock score of −152.823, and RMSD value of 1.01 Å, suggesting that the applied docking protocol was highly preferred.

#### 2.6.4. Phosphate Acetyltransferase (Pta)

Phosphate acetyltransferase (Pta) plays an important role in acetate metabolism along with acetate kinase. This enzyme catalyzes the uptake of carbohydrates and their conversion into their respective phosphoesters during transport [80]. It is also involved in other metabolic pathways such as taurine and hypotaurine metabolism, pyruvate metabolism, and propanoate metabolism [81]. It was reported that this enzyme activity is important for virulence in pathogenic bacteria including *S. saprophyticus* [82]. Therefore, it is essential for the survival of bacteria and could be a putative drug target for the design and evaluation of a new class of antimicrobials [80]. In the template crystal structure of Pta protein (PDB ID: 4E4R from *S. aureus* subsp. *aureus* MRSA252), the native crystallographic ligand 2-amino-2-hydroxymethyl-propane-1,3-diol (TRS) created H-bonds in the active site residues, Gly130 and Asp305. However, none of these residues were involved in the H-bond formation with the ZINC4270981 compound. Instead, this compound formed two H-bonds with the other active site residues, Gln325, and Leu299 of Pta protein with a predicted MolDock score of −134.847 (Figure 5a,b). The redocking of native ligand (TRS) with Pta (PDB ID: 4E4R) revealed the MolDock score of −38.370, and RMSD value of −1.12 Å, indicating that the applied protocol was fair for this protein.

#### 2.6.5. UDP-N-Acetylmuramoyl Alanyl-D-Glutamate-2,6-diaminopimelate Ligase (MurE)

The target enzyme MurE ligase is a complex molecule that initiates the peptidoglycan biosynthesis [83] by adding meso-diaminopimelic acid to the nucleotide precursor UDP-N-acetylmuramoyl-L-alanyl-D-glutamate during the synthesis of murein in the cytoplasm [84]. As this enzyme is vital to the survivability of *S. aureus* strains, it can be used as a potential antibacterial drug target [57,58,83]. Based on the interaction between the crystallographic structure of the MurE template (4C12 from *S. aureus*) and the crystallographic ligand uridine 5′ diphospho N-acetyl muramoyl-L-Alanyl-D-Glutamyl-L-Lysine (UML), it was found that the active site residues involved in H-bond interactions were Ser456, Glu460, Asp406, Thr152, Ser179, Arg187, Arg383, His205, Asn151, Thr153, Thr45, Thr46, Val47, Thr28, Ser30, and Val47. However, the ZINC4235426 compound created four H-bonds with the active site residues, Tyr45, Thr46, Val47, and Glu155 of MurE ligase with a MolDock score of −125.654 (Figure 6a,b). 

The redocking of the native ligand (UML) with MurE (PDB ID: 4C12) generated the MolDock score of −167.754 and RMSD value of 4.75 Å. The RMSD value of redocking found above the cutoff value (<2 Å) could be the reason for the large ligand size (120 atoms), and more rotatable bond (26 flexible torsions). This finding was supported by the earlier study, which suggested that redocking accuracy decreases as the number of rotatable bonds increases, regardless of the docking program used [85]. Also, the typical 2 Å RMSD cutoff for docking accuracy may not be reliable for the ligands with a large size and number of rotatable bonds [86].

#### 2.6.6. UTP-Glucose-1-Phosphate Uridylyltransferase (UGPase)

This enzyme, also known as UDP-glucose pyrophosphorylase or UGPase, is ubiquitous due to its indispensable roles in glycogen synthesis and production of glycolipids, glycoproteins, and proteoglycans [87,88]. It is also required for the capsular polysaccharide biosynthesis and is a determinative virulence factor in *Streptococcus pneumoniae* [89]. A defective UGPase fails to incorporate galactose into its cell wall [89]. The indispensability of the UGPase projects the enzyme as a potential drug target [90]. In the template crystal structure of UGPase (PDB ID: 5VCT from *Burkholderia ambifaria* MC40-6), the native crystallographic ligand citric acid (CIT) created two H-bonds in the active site residues, Lys16 and Leu14. However, in the case of the ZINC428871 compound, none of these residues were involved in the H-bond formation with compounds. Instead, the ZINC428871 compound formed a single H-bond with the other active site residues, Leu110 of UGPase, with a predicted MolDock score of −122.664 (Figure 7a,b). The redocking of native ligand (CIT) with UGPase (PDB ID: 5VCT) achieved a MolDock score of 29.116, and RMSD value of 1.84, which suggested that the applied docking simulation protocol was satisfied for this protein.

#### 2.6.7. Putative Fatty Acid Synthesis Protein (PlsX)

Putative fatty acid synthesis protein (PlsX) is the key enzyme that coordinates the fatty acid synthase II (FASII) pathway to the phospholipid synthesis pathway and allows passage of unsaturated fatty acids into the membrane [91]. The FASII pathway is used by bacteria to produce the fatty acid components of phospholipids essential in human pathogens [92]. The essential role of fatty acid in membrane structure integrity has provided attention on targeting this pathway [93]. The B-chain structure of the 1U7N template from *Enterococcus faecalis* for PlsX has no native ligand. Therefore, cavity 1 (V:180.224 Å; S:660.48 Å, R:19 Å) identified by the MVD tool was selected for docking with the ZINC4237105 compound. This compound formed one H-bond to the active site residue, Lys262 of PlsX protein, with a MolDock score of −130.756 (Figure 8a,b).

#### 2.6.8. Pantoate Beta-Alanine Ligase (PanC)

Pantoate beta-alanine ligase (PanC) is the last enzyme involved in pantothenate biosynthesis. It catalyzes the adenosine triphosphate (ATP)-dependent condensation of pantoate and β-alanine to form pantothenate (vitamin B5) [94]. The essentiality of PanC for the growth and survival of *S. aureus* can be gauged by the fact that the enzyme is central to fatty acid metabolism [95]. Based on a structural comparison with a crystallographic structure of PanC template (PDB ID: 3AG6 from *S. aureus* subsp. *aureus* NCTC 8325), the active site residues involved in H-bonds with the crystallographic native ligand pantoyl adenylate (PAJ) were Gln154, Gln62, Met31, Gly148, His35, His38, Lys185, and Val177. The ZINC4235426 compound created four H-bonds with the active site residues, Met31, Gly148, His35, and Thr30 of PanC protein, with a MolDock score of −173.843 (Figure 9a,b). The redocking of native ligand (PAJ) with PanC (PDB ID: 3AG6) generated the MolDock score of −174.601, and the RMSD value of −1.57 Å, indicating that the used protocol for docking simulation was highly preferable.

The Swiss ADME analysis results showed that all the drug molecules listed in Table 2 satisfy Lipinski’s rule of five with zero violations and had no PAIN alerts. Additionally, ZINC4235426, ZINC4259578, and ZINC428871 assured the properties of drug-likeness (Ghose, Veber, Egan, and Muegge). The *K*p values of the drug molecules are in the ranges of −6.21 to −7.32 cm/s, suggesting low skin permeability [65]. The ZINC4235924 and ZINC4270981 molecules were found to have high blood–brain barrier (BBB) permeability. All the compounds are substrates of permeability glycoprotein (P-gp) except ZINC428871. The logP values are predicted in the ranges of 0.82 to 4, indicating that the drug molecules have optimal lipophilicity. The drug molecules showed high gastrointestinal (GI) absorption and bioavailability scores, and have non-carcinogenicity.

## 3. Materials and Methods

### 3.1. Genome Sequences

The genome sequences of sixteen MRSA strains belonging to community-associated and hospital-associated strains with the most predominant clonal complexes were retrieved from the NCBI database (http://www.ncbi.nlm.nih.gov/genbank/; accessed on 25 July 2020). The genomes were re-annotated using RAST (Rapid Annotation using Subsystem Technology) platform [96] to avoid unexpected and inappropriate gene interpretation results. The features of these MRSA strains were summarized in Appendix A.

### 3.2. Prediction of Core Proteome

The identification of conserved proteins (core proteome) among the sixteen genomes was analyzed using reciprocal best BLAST hits of all CDS in EDGAR v.2.0 software framework [97]. In this analysis, the genome of *S. aureus* subsp. *aureus* HO 5096 0412 (HE681097.1) was used as a reference strain, and the rest of all the MRSA genomes were compared with the reference strain. Only those proteins that were common in all the strains were selected as the core proteome.

### 3.3. Identification of Essential Non-Homologous Proteins

Paralogs or redundant sequences from the MRSA core proteome were removed using the CD-HIT module with sequence identity cutoff of 0.7 (70%) in CD-HIT suite [98]. CD-HIT suite is widely used for comparing and clustering proteins that satisfy the sequence identity cutoff and generate one representative sequence of each cluster as well as a list of clusters. Further core essential proteins of *S. aureus* were analyzed using GEPTOP 2.0 with an essentiality score cutoff of 0.24 [99]. Geptop provides a platform for the identification of essential genes for prokaryotic organisms by comparing the orthology and phylogeny of query proteins against the datasets defined experimentally in the database of essential genes (DEG). The essential non-paralogous protein sequences were subjected to BLASTp against the genome of *Homo sapiens* [100] using default parameters. The resultant sequences showing significant similarity with the Human host were discarded, while non-homologous sequences with no hit were selected for subsequent analysis.

### 3.4. Characterization and Prediction of Subcellular Localization

The essential non-host homologous protein sequences were used for the prediction of drug targets and vaccine candidates. The non-host homologous protein sequences were used for the prediction of subcellular location using optimized PSORTb 3.0 [101] and CELLO (subCELlular LOcalization predictor) v.2.5 [102]; PSORTb and CELLO servers using a support-vector machine (SVM)-based method to predict the subcellular localization of proteins as cytoplasmic (CYT), secreted (SEC), potentially surface-exposed (PSE), and membrane (MEM).

### 3.5. Reverse Vaccinology Approach for Prediction of Putative Vaccine Candidates

The proteins localized in the extracellular or outer membrane regions of the organism are highly important for reverse vaccinology since these proteins are the first to be in contact with host immune cells and stimulate immune responses [103]. The secreted (SEC), potentially surface-exposed (PSE), and membrane (MEM) protein sequences were submitted to the Vaxign v.2.0 tool [104] to predict the major histocompatibility complex (MHC class I and II) binding properties with adhesion probability greater than 0.5, the number of transmembrane helices, and no similarity to host proteins. Vaxign used different computational programs such as pSORT for subcellular localization, HMMTOP for transmembrane helices prediction, SPAAN for adhesin probability identification, OrthoMCL for the detection of human host orthologue proteins, and Vaxitope for prediction of MHC-I- and MHC-II-binding epitopes present in the input protein sequence [104]. B-cell epitopes were predicted using SVMTriP with epitope length 20 amino acid (aa) parameter [105]. The antigenic epitope prediction approach by SVMTriP is based on the amino acid properties (hydrophilicity, solvent accessibility, secondary structure, flexibility, and antigenicity). This tool utilizes Support Vector Machine (SVM) by combining the Tri-peptide similarity and Propensity scores (SVMTriP) for better prediction performance. The antigenicity of vaccine candidate proteins was predicted using the VaxiJen v.2.0 server with a threshold of 0.4 [64], and ANTIGENpro (http://scratch.proteomics.ics.uci.edu/; accessed on 8 March 2022) [106]. VaxiJen employed auto and cross-covariance (ACC) transformation of protein sequences into uniform vectors of major amino acid properties to evaluate the antigenicity [64]. Allergenicity of vaccine candidate proteins was identified using AllerTOP v.2.0, an online server (http://www.ddg-pharm fac.net/AllerTOP; accessed on 5 February 2022). AllerTOP utilizes the k-nearest neighbors (kNN), auto- and cross-covariance (ACC) transformation, and amino acid E-descriptors machine learning techniques for the classification of allergens by exploring the physiochemical properties of proteins. The instability index and molecular weight (mol. wt) of the vaccine candidate proteins were analyzed using the ProtParam tool [100]. The sequence-based domain information of vaccine candidates was retrieved from the conserved domain database, CDD (https://www.ncbi.nlm.nih.gov/Structure/cdd/wrpsb.cgi, accessed on 8 January 2022) [107].

### 3.6. High-Throughput Structural Modelling

The essential non-host homologous cytoplasmic (CYT) protein sequences were subjected to analyze for potential drug targets, since these proteins play a role in the basic survival processes of the organism. These CYT proteins avoid cross-binding of drugs with human proteins, and drug side effect probability [108]. These CYT proteins were submitted to the MHOLline 2.0 server [109] to model three-dimensional (3D) structures of cytoplasmic proteins. This software integrates with HMMTOP, BLAST, BATS, MODELLER, and PROCHECK to analyze and classify potential drug targets based on their structural quality. The submitted protein sequences detected the presence of transmembrane regions by the HMMTOP program, and template structure against the Protein Data Bank by BLAST algorithm. Further, the BATS (Blast Automatic Targeting for Structures) tool conducts refinements in the template search, which is a key step for the model construction. The BATS program generates the BLAST output files into G0, G1, G2, and G3, in which G0 represent non-aligned sequence, G1 indicate E > 10 × 10^−5^ or identity < 15%, G2 signify E ≤ 10 × 10^−5^, identity ≥ 25%, and length variation index (LVI) ≤ 0.7, G3 imply E ≤ 10 × 10^−5^, identity ≤ 15% to <25% or LVI > 0.7. The G2 group was selected for the 3D model construction and generates the files for automated alignment used by the MODELLER program. The generated 3D model was evaluated based on their stereochemical quality by the PROCHECK program. Also, MHOLline 2.0 server provides information on the 3D model, a Ramachandran plot, structural quality, and enzymatic function. 

### 3.7. Druggability Analysis of Drug Targets

The drug target proteins from the G2 group belonged to very-high-, high- and good-quality proteins that were analyzed for druggability using DoGSiteScorer [110], an automated pocket detection and analysis tool for calculating the druggability of protein cavities. For each detected cavity, the tool returns the pocket residues and a druggability score ranging from 0 to 1. The druggable scores closer to 1 are more druggable and likely to bind ligands with high affinity [110]. The druggability cavity of each drug target with a druggable score greater than 0.7 was selected for the docking analysis. Also, to determine the potential drug targets, the molecular weight of protein with less than 100 kDa was considered as a therapeutic target, suggesting that these proteins have a possibility to experimentally research for drug development [111]. The molecular weight (mol. wt) of the target proteins was calculated using the ProtParam tool [100]. We chose the eight best drug targets based on the above parameters and further analyzed them for molecular function (MF) and biological process (BP) using UniProt [112].

### 3.8. Ligand Libraries and Docking Analyses

The ligands, drug-like molecules (Natural Product and its derivatives) described earlier by [113] were obtained from the ZINC 15 database [114] and the in-house library was constructed. The 3D structures of all the target proteins were inspected for structural errors such as wrong bonds, missing atoms, and protonation states through Molegro Virtual Docker (MVD) 6.0 [68]. The cavities generated by MVD for each target were compared with the cavities detected by DoGSiteScorer (druggability ≥ 0.80). The most druggable cavity defined by MVD was subjected to virtual screening. The MolDock Optimizer search algorithm was used in this analysis, which is based on a differential evolutionary algorithm, using the default parameters, that are (a) population size = 50; (b) scaling factor = 0.5; (c) crossover rate = 0.9. The 3D poses and 2D representation of docked molecules were analyzed in PyMOL 2.4.1 [115] and Discovery Studio Visualizer 2020 [116].

## 4. Conclusions

Methicillin-resistant *S. aureus* is the leading cause of nosocomial and community infections. The genome plasticity of *S. aureus* facilitates the emergence of hypervirulent strains and poses a greater threat to public health. To promote long-term protection against multidrug-resistant (MDR) *S. aureus* infections, there is an urgent need for the development of novel vaccine candidates or drug target proteins. In the present study, reverse vaccinology and subtractive genome approaches were applied to predict potential vaccine/drug targets, which can be used in the development of promising vaccines and drugs for MDR *S. aureus*. The predicated vaccine candidate proteins were based on diligent analysis of protein sequences and different immune databases. These studies reduce time, labor, and costs for researchers, and finding results could be for desired solutions to prevent MRSA infections. A reverse vaccinology approach recommended two proteins (PrsA, and EssA) that have a high potential for vaccine candidates. A molecular docking approach of selected drug target proteins with the drug-like molecules revealed that the ZINC4235426 is a potential drug molecule with favorable interactions and low MolDock Scores with the target active site residues of 5 drug target proteins *viz.,* biotin protein ligase, HPr kinase/phosphorylase, thymidylate kinase, UDP-N-acetylmuramoyl-L-alanyl-D-glutamate-L-lysine ligase, and pantothenate synthetase. Thus, the identification of molecules in our current in silico study could be potentially used as new drugs for the treatment of MRSA infections. The identified proteins can be used for further rational drug or vaccine design to identify novel therapeutic agents for the treatment of MDR staphylococcal infection. Further, the identified proteins through in silico approaches are required to be validated using in vitro and in vivo experiments to find new methods of treating MRSA strain-induced life-threatening diseases.

## Figures and Tables

**Figure 1 molecules-27-02083-f001:**
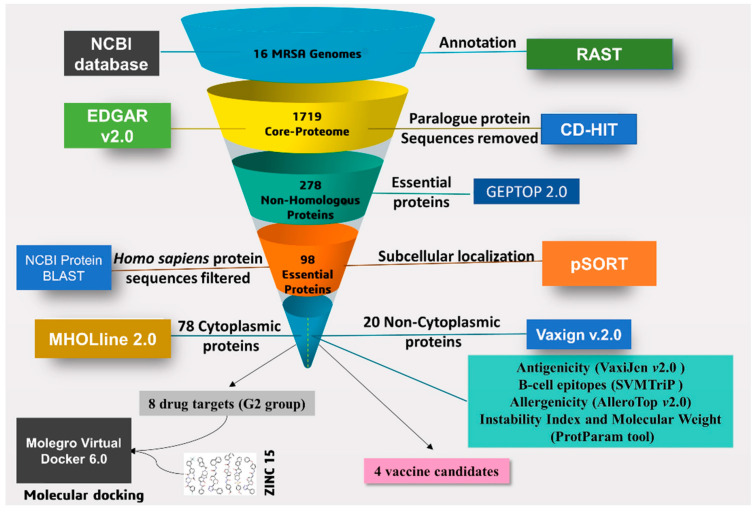
Systemic workflow of vaccine and drug targets identification using subtractive genome analysis.

**Figure 2 molecules-27-02083-f002:**
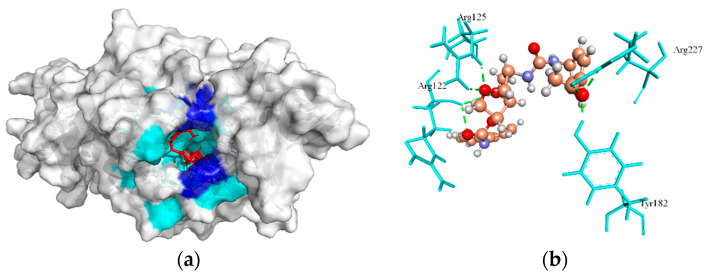
The molecular docking analysis of BPL (WP_000049913.1) with compound ZINC4235426. (**a**) 3D surface representation of ZINC4235426 (red) and BPL interactions with hydrogen bonding sites (blue), and hydrophobic interactions (cyan). (**b**) Residues (cyan) involved in the H-bond interaction (green dashed lines) with the compound (scaled ball and stick).

**Figure 3 molecules-27-02083-f003:**
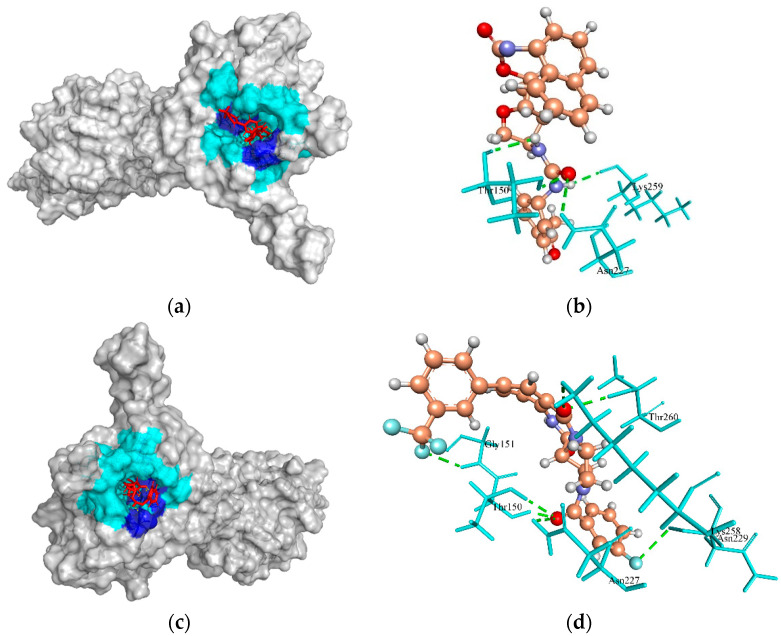
The molecular docking analysis of HPr kinase (WP_000958224.1) with compounds ZINC4235426 and ZINC4235924, respectively. (**a**,**c**) 3D surface representation of ZINC4235426 (red), ZINC4235924 (red), and HPr kinase interactions with hydrogen bonding sites (blue), and hydrophobic interactions (cyan). (**b**,**d**) Residues (cyan) involved in the H-bond interaction (green dashed lines) with the compound (scaled ball and stick).

**Figure 4 molecules-27-02083-f004:**
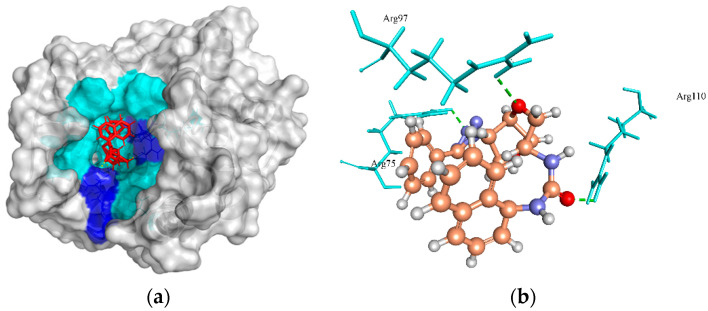
The molecular docking analysis of TMK (WP_001272126.1) with compound ZINC4259578. (**a**) 3D surface representation of ZINC4259578 (red) and TMK interactions with hydrogen bonding sites (blue), and hydrophobic interactions (cyan). (**b**) Residues (cyan) involved in the H-bond interaction (green dashed lines) with the compound (scaled ball and stick).

**Figure 5 molecules-27-02083-f005:**
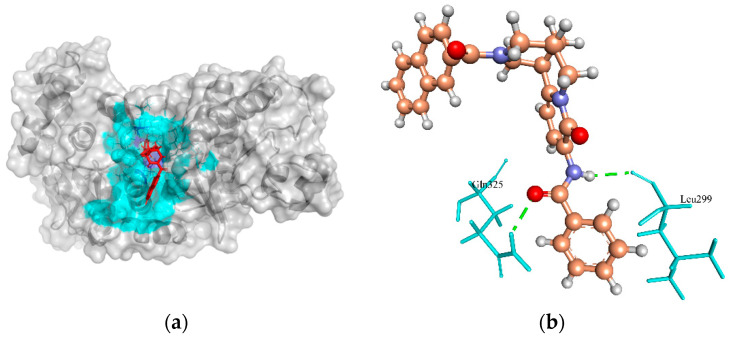
The molecular docking analysis of Pta (WP_000774281.1) with compound ZINC4270981. (**a**) 3D surface representation of ZINC4270981 (red) and Pta interactions with hydrogen bonding sites (blue), and hydrophobic interactions (cyan). (**b**) Residues (cyan) involved in the H-bond interaction (green dashed lines) with the compound (scaled ball and stick).

**Figure 6 molecules-27-02083-f006:**
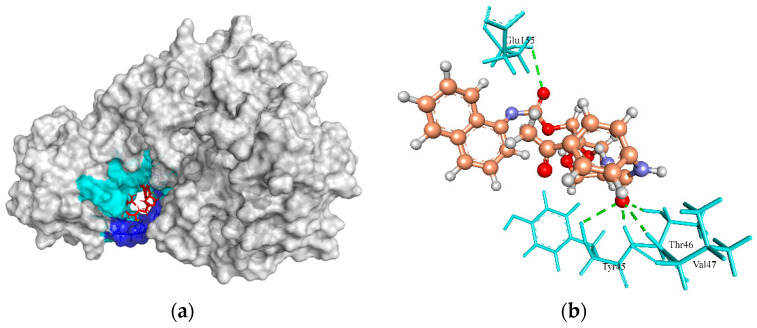
The molecular docking analysis of MurE (WP_000340119.1) with compound ZINC4235426. (**a**) 3D surface representation of ZINC4235426 (red) and MurE interactions with hydrogen bonding sites (blue), and hydrophobic interactions (cyan). (**b**) Residues (cyan) involved in the H-bond interaction (green dashed lines) with the compound (scaled ball and stick).

**Figure 7 molecules-27-02083-f007:**
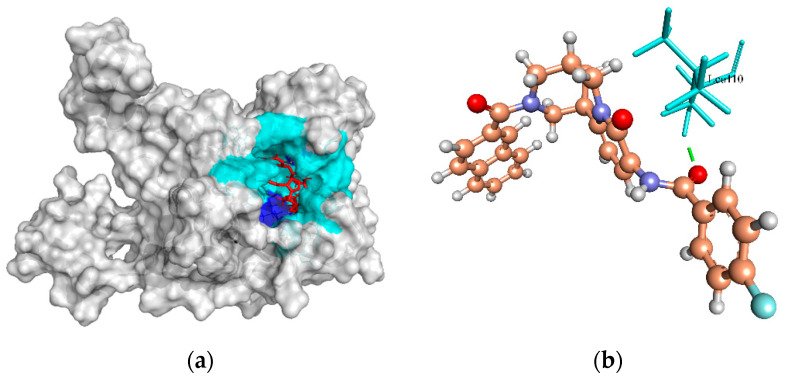
The molecular docking analysis of UGPase (WP_000721337.1) with compound ZINC428871. (**a**) 3D surface representation of ZINC428871 (red) and MurE interactions with hydrogen bonding sites (blue), and hydrophobic interactions (cyan). (**b**) Residue (cyan) involved in the H-bond interaction (green dashed lines) with the compound (scaled ball and stick).

**Figure 8 molecules-27-02083-f008:**
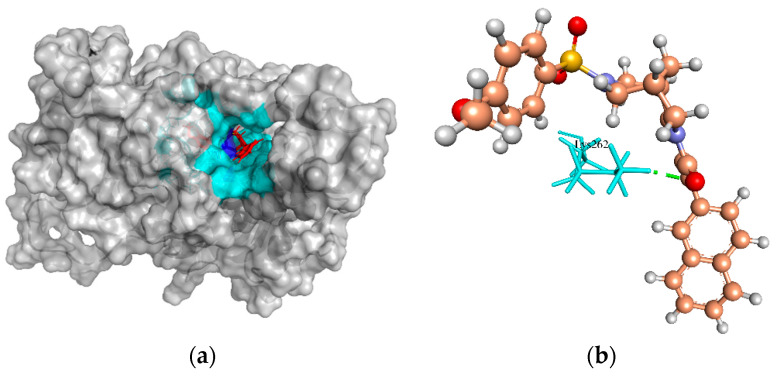
The molecular docking analysis of PlsX (WP_000239744.1) with compound ZINC4237105. (**a**) 3D surface representation of ZINC4237105 (red) and PlsX interactions with hydrogen bonding sites (blue), and hydrophobic interactions (cyan). (**b**) Residue (cyan) involved in the H-bond interaction (green dashed lines) with the compound (scaled ball and stick).

**Figure 9 molecules-27-02083-f009:**
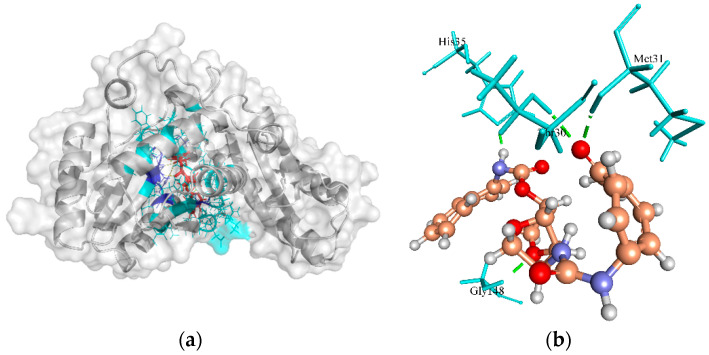
The molecular docking analysis of PanC (WP_000163742.1) with compound ZINC4235426. (**a**) 3D surface representation of ZINC4235426 (red) and PanC interactions with hydrogen bonding sites (blue), and hydrophobic interactions (cyan). (**b**) Residues (cyan) involved in the H-bond interaction (green dashed lines) with the compound (scaled ball and stick).

**Table 1 molecules-27-02083-t001:** Vaccine target candidates for methicillin-resistant *S. aureus* identified by Vaxign v.2.0.

Vaccine Target	Protein ID *	Length (bp)	Mol. Wt. (kDa)	Adherence Score	B-Cell Epitope Peptides	VaxiJen Score	Trans-Membrane Helix	Allergen
Foldase protein (PrsA)	WP_000782119.1	320	35.623	0.68	5	0.7662	0	No
ESAT-6 machinery protein (EssA)	WP_000928935.1	152	17.392	0.58	1	0.7034	1	No
Penicillin-binding protein 1 (PBP1)	WP_001118663.1	744	82.738	0.53	10	0.6351	1	No
DD-transpeptidase (PBP2)	WP_000138351.1	727	80.356	0.82	10	0.6846	1	No

Protein ID * denotes NCBI protein accession ID.

**Table 2 molecules-27-02083-t002:** Docking studies of drug-like molecules (ZINC compounds) with eight drug target proteins. The table shows the MolDock score, number of hydrogen bonds, and active site residues of target proteins with the respective ZINC compounds.

Target Proteins	Position (x, y, z)	ZINC ID	MolDock Score	H-Bonds/Residues
Biotin protein ligase	53.60, 18.59, 18.89	ZINC4235426	−176.846	7/Tyr182, Arg227, Arg125, and Arg122
HPr kinase/phosphorylase	−60.93, 127.77, −83.27	ZINC4235426	−147.451	4/Lys259, Thr150, and Asn227
		ZINC4235924	−137.549	7/Gly151, Thr150, Asn227, Lys258, Thr260, and Asn229
Thymidylate kinase	5.82, 10.91, 16.59	ZINC4259578	−139.656	3/Arg75, Arg97, and Arg110
		ZINC4235426	−139.150	6/Arg75, Arg97, Glu106, Tyr105, and Glu42
Phosphate acetyltransferase	17.38, 20.45, 53.06	ZINC4270981	−134.847	2/Gln325, and Leu299
UDP-N-acetylmuramoyl-L-alanyl-D-glutamate-L-lysine ligase	99.73, 29.35, 28.72	ZINC4235426	−125.654	4/Tyr45, Thr46, Val47, and Glu155
UTP-glucose-1-phosphate uridylyltransferase	−19.38, −1,2.96, −16.82	ZINC428871	−122.664	1/Leu110
Fatty acid/phospholipid synthesis	38.55, 51.23, 49.04	ZINC4237105	−130.756	1/Lys262
Pantothenate synthetase	−6.64, −1.01, −43.92	ZINC4235426	−173.843	4/Met31, Gly148, His35, and Thr30

## Data Availability

The following information was supplied regarding data availability: The *S. aureus* genome sequences used in this study are available at https://www.ncbi.nlm.nih.gov/nucleotide/ (accessed on 20 January 2022) under the accession number given in Appendix A.

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
