# Peer review of "Identification of Putative Vaccine and Drug Targets against the Methicillin-Resistant Staphylococcus aureus by Reverse Vaccinology and Subtractive Genomics Approaches"

_molecules, 2022, doi:10.3390/molecules27072083_

Round 1

Reviewer 1 Report

The authors used in silico approaches to detect significant proteins for developing potential vaccines and drug targets using reverse vaccinology and subtractive genome approaches. It seems that the final proteins assumed as suitable for vaccine development and drug targets are already known and studied for their properties. In the final conclusion the authors propose only one molecule as suitable potential drug candidate but no drug targets as mentioned in the title. The predictions of this drug candidate rely on docking protocols, which were not validated. Also questions arise about the defining of the binding sites (BS) - especially when there are available crystallographic structures of proteins in complexes with natural ligands. It is unclear if the detected BS is the same with the one from crystallographic data. It is unclear how this final molecule is assumed as the most promising drug candidate. The selection criteria of the protein drug targets must also be clarified.

Additionally the introduction must be improved and include the significance of the research area, the worldwide spread of the issue, the current situation with drugs, vaccines and drug targets.

A lot of questions could be raised about the used methods/servers and methodology. This part should be explained in more details.

The number of the initial set of molecules form ZINC 15 database is unclear as well as the criteria for set selection.  

For these reasons I could not recommend the manuscript for publication in Molecules.

Author Response

Responses to Reviewer 1

Many thanks for your positive comments and for calling our attention to improve the manuscript

Comments and Suggestions for Authors

The authors used in silico approaches to detect significant proteins for developing potential vaccines and drug targets using reverse vaccinology and subtractive genome approaches. It seems that the final proteins assumed as suitable for vaccine development and drug targets are already known and studied for their properties. In the final conclusion the authors propose only one molecule as suitable potential drug candidate but no drug targets as mentioned in the title. The predictions of this drug candidate rely on docking protocols, which were not validated. Also questions arise about the defining of the binding sites (BS) - especially when there are available crystallographic structures of proteins in complexes with natural ligands. It is unclear if the detected BS is the same with the one from crystallographic data. It is unclear how this final molecule is assumed as the most promising drug candidate. The selection criteria of the protein drug targets must also be clarified.

Response: Thank you for your valuable suggestion. The final drug target and other target proteins docked with the suggested drug molecules have not been reported yet. The promising drug molecule showed interaction not only to the final drug target but also interacted with the other drug target proteins. The conclusion has been revised as per your suggestion. We agreed that the predictions of the drug candidate rely on molecular docking. We are grateful for your comment, but due to the insufficient resources and funds for validation, this work could not be completed for the current manuscript. In the future, this work will be continued after getting sufficient funds and resources for biological validation. In the case of defining the binding site residues of target proteins, the DoGSiteScorer tool was used to detect the binding site residues or druggable region/cavity, and further checked the cavity using  Molegro Virtual Docker. Those defined cavities were used for molecular docking. In the case of some proteins, the cavity defined with native ligand had poor volume, surface area, and druggable score. As a result, the mentioned tools defined the large region/cavity that had higher druggability. To make it clearer, the detected BS residue of target proteins that were involved in the interactions has been incorporated in Supplementary Table S5. The final drug molecule is assumed to be a promising one among the others because this drug molecule showed better interaction with most of the identified drug target proteins with the lowest MolDock scores. Further, this drug molecule obeyed Lipinski’s rule of 5 and other ADME parameters. Also, as per your suggestion, the selection criteria of the protein drug target have been incorporated in lines numbers 146-164.

Additionally the introduction must be improved and include the significance of the research area, the worldwide spread of the issue, the current situation with drugs, vaccines and drug targets.

Response: Thank you for your kind suggestion. As per your suggestion, the ‘Introduction’ has been revised.

A lot of questions could be raised about the used methods/servers and methodology. This part should be explained in more details.

Response: Thank you for your valuable suggestion. As suggested, the methodology parts have been explained in more detail.

The number of the initial set of molecules from ZINC 15 database is unclear as well as the criteria for set selection.  

Response: As per your suggestion, this part has been updated in section 2.8.

Reviewer 2 Report

The study by Romen singh on “Identification of putative vaccine and drug targets against the methicillin-resistant Staphylococcus aureus by using reverse vaccinology and subtractive genomics approaches is nicely executed but needs improvement before publication. In this study authors study to identify vaccine candidates and drug target proteins against the 16 strains of MRSA using molecular docking approach of selected eight drug target proteins with the drug-like molecules and identified various target active site residues like biotin protein ligase, HPr kinase/phosphorylase, thymidylate kinase, UDP-N-acetylmuramoyl-L-alanyl-D-glutamate-L-lysine ligase, and pantothenate synthetase. The study is overall good but needs some edits and further clarifications. Following are the points which authors need to address

  1. Authors need to address/elaborate CD-HIT and GEPTOP programs which they have used to remove paralogs or redundant sequences from the MRSA core-proteome Authors should give the details in supplementary about nonhomologous sequences which they used for further analysis.
  2. The non-host homologous protein sequences were further used for the prediction of subcellular location using optimized PSORTb 3.0 and assembled in the Vaxign v.2.0 tool. This methodology is not convincing. Are there other programs which authors can check.
  3. Authors should provide the sequence of surface exposed (PSE), and membrane (MEM) protein sequences
  4. Did authors use other parameters to identify the antigenicity of vaccine candidate proteins apart from VaxiJen v.2.0 server with a threshold of 0.4
  5. Authors need to explain why the molecular weight of proteins less than 100 kDa were considered as druggable molecules

Author Response

Responses to Reviewer 2

Many thanks for your positive comments and for calling our attention to improve the manuscript

Comments and Suggestions for Authors

The study by Romen singh on “Identification of putative vaccine and drug targets against the methicillin-resistant Staphylococcus aureus by using reverse vaccinology and subtractive genomics approaches is nicely executed but needs improvement before publication. In this study authors study to identify vaccine candidates and drug target proteins against the 16 strains of MRSA using molecular docking approach of selected eight drug target proteins with the drug-like molecules and identified various target active site residues like biotin protein ligase, HPr kinase/phosphorylase, thymidylate kinase, UDP-N-acetylmuramoyl-L-alanyl-D-glutamate-L-lysine ligase, and pantothenate synthetase. The study is overall good but needs some edits and further clarifications. Following are the points which authors need to address

  1. Authors need to address/elaborate CD-HIT and GEPTOP programs which they have used to remove paralogs or redundant sequences from the MRSA core-proteome Authors should give the details in supplementary about nonhomologous sequences which they used for further analysis.

Response: Thank you for your suggestion. As per your suggestion, CD-HIT and GEPTOP programs descriptions have been incorporated in lines numbers 96-101. Non-homologous sequences have been provided in the supplementary section (Sequence file 1).

  1. The non-host homologous protein sequences were further used for the prediction of subcellular location using optimized PSORTb 3.0 and assembled in the Vaxign v.2.0 tool. This methodology is not convincing. Are there other programs which authors can check.

Response: Thank you for your valuable suggestion. As suggested, we have further performed subcellular location using CELLO v.2.5.

  1. Authors should provide the sequence of surface exposed (PSE), and membrane (MEM) protein sequences

Response: As per your suggestion, the sequences of surface-exposed (PSE), and membrane (MEM) proteins have been provided in the supplementary section (Sequence file 2)

  1. Did authors use other parameters to identify the antigenicity of vaccine candidate proteins apart from VaxiJen v.2.0 server with a threshold of 0.4

Response: As suggested, we further analyzed the antigenicity of vaccine candidate proteins using the ANTIGENPro tool.

  1. Authors need to explain why the molecular weight of proteins less than 100 kDa were considered as druggable molecules

Response: As per your suggestion, the sentence has been revised in line numbers 173-175.

Reviewer 3 Report

My comments;

This manuscript is interesting and also well designed. This paper is based on rigorous academic standards.

  1. The research methodology for the study is appropriate .
  2. The supporting evidence in this paper is strongly reliable and properly validated.
  3. The results of analysis are correctly interpreted .

Author Response

Thank you for your positive comments.

Round 2

Reviewer 1 Report

The manuscript is significantly improved and clarified. However there are few issues to be considered:

A control, in the docking calculations need to be included in the cases of BPL, TMK, Pta, MurE, UGPase and PanC. This means that the original ligand from the crystallographic data should be re-docked under the same docking protocol. The corresponding MolDock scores and RMSD values should be included in the Results and Discussion section.

Are there any PAINS alerts predicted by the Swissadme server for the discussed ligands?

How many ligands were selected from ZINC15 database for docking?

Vaxijen = VaxiJen in lines 271 and 341.
